# Psychological distress, employment, and family functioning during the COVID-19 outbreak among recent immigrant families in Israel: Moderating roles of COVID-19 prevalence

Tae Kyoung Lee[1]*, Maya Benish-Weisman[2], Saskia R. Vos[3], Maria Fernanda Garcia[4], Maria C. Duque Marquez[5], Ivonne A. Calderón[3], Tatiana Konshina[2], Einat Elizarov[6], Seth J. Schwartz[5]

1 Department of Child Psychology and Education / Convergence for Social Innovation, Sungkyunkwan University, Seoul, Republic of Korea, 2 Paul Baerwald School of Social Work and Social Welfare, Hebrew University of Jerusalem, Jerusalem, Israel, 3 Department of Public Health Sciences, University of Miami Miller School of Medicine, Miami, Florida, United States of America, 4 Department of Community Well-being, University of Miami, Miami, FL, United States of America, 5 Department of Educational Psychology, University of Texas, Austin, TX, United States of America, 6 Department of Counseling and Human Development, University of Haifa, Haifa, Israel

* ltk501@skku.edu

## Abstract

Grounded in an ecodevelopment perspective, in the current study we examined unique and moderating effects of daily COVID-19 prevalence (social contexts) on effects of COVID-19 related risk and protective factors such as emotional distress (individual contexts) and employment (working from home and unemployment status; family contexts) on family functioning among 160 recent immigrant families in Israel. In general, results indicate several unique effects of COVID-19 related factors (such as COVID-19 emotional distress, unemployment, and remote work arrangements) on both parents' and adolescents' reports of family functioning. However, results indicated that there were more significant associations between COVID-19 factors (e.g., emotional distress and COVID-19 prevalence) and family functioning indicators with adolescents, than with parents. The effects of COVID-19 factors (e.g., emotional distress and remote work arrangements) were moderated by daily COVID-19 prevalence (new cases and deaths). We discuss ways in which interventionists can contribute to pandemic-related research to promote optimal family functioning among immigrant families.

## Introduction

The coronavirus (COVID-19) pandemic began in December 2019 and spread globally, causing countries to shut down the majority of international travel, impose social distancing guidelines, discourage most social gatherings, and shut down much of their economies. The

**Data Availability Statement:** All relevant data are within the manuscript and its Supporting Information files.

**Funding:** This project is supported by the US-Israel Binational Science Foundation (BSF) (2018091). Any opinions, findings, conclusions, or recommendations expressed in this material are those of the authors and do not necessarily reflect the views of the BSF. The funders had no role in study design, data collection and analysis, decision to publish, or preparation of the manuscript.

**Competing interests:** The authors have declared that no competing interests exist.

pandemic has distanced people from their workplaces, schools, and friends, while forcing families to shelter in place. These precautions have had led to prolonged social isolation and increased time spent at home with family members, at the same time as financial strains for individuals and families continue to soar. Prevention scientists are therefore concerned about the potential effects of COVID-19 and associated mitigation strategies on family functioning (e.g., family conflict, decreased positive parenting) [1], where family functioning is closely linked to family members' mental and behavioral health. Some studies have begun to examine the impacts of COVID-19-related risks on family functioning among families in the U.S. [2] and Europe [3, 4]. For example, a recent study indicated that Italian families reported high levels of stress and tension during the COVID-19 pandemic [4]. However, less is known about COVID-19 effects on families in other countries outside the U.S. and Europe.

Israel is an interesting case study for learning about the psychological and domestic effects of COVID-19. Starting in March 2020, Israel declared lockdown almost immediately after the start of the outbreak. Israel initially reported one of lowest COVID-19 prevalence rates [5] with roughly 2,000 active cases, less than 300 deaths, and a fatality rate of 1.67%, substantially lower than that of most Western countries and well below the world average of 6.15% [6]. However, starting in June 2020, positive cases in Israel rapidly increased. In September 2020, Israel had one of the world's highest rates of per capita coronavirus cases and deaths [7]. By September 2020, Israel's daily death rate was 3.5 deaths per million people, whereas the US rate was 2.2 deaths per million people [8].

These rapid increases in COVID-19 prevalence (new cases and deaths) within a relatively short period of time may have both positively and negatively influences Israeli family dynamics. For example, as COVID-19 prevalence increased, many workplaces and schools shifted to virtual work and learning from home. These arrangements served to increase the amount of time that family members spend together, creating more options to communicate, which in turn may positively influenced family interactions and perceptions of family functioning among parents and their children [9]. However, the COVID-19 pandemic also created family economic hardship, compromising parents' (or other household adults') employment status (e.g., job loss), which may negatively influence family functioning. Additionally, as an unpredictable illness whose effects can range from mild symptoms to difficulty breathing and cardiac problems, COVID-19 is known to be associated with worry and panic (e.g., COVID-19 emotional distress; hereafter referred as to "COVID-19 distress") among family members [10]. These adverse effects of COVID-19 (e.g., unemployment status and COVID-19 distress) may negatively influence family functioning. Therefore, it is important to understand the effects of COVID-19 on family functioning among Israeli families.

In the present study we focused on recent immigrant families from the Former Soviet Union (FSU) in Israel, as they are the largest immigrant group in the country. In total, FSU immigrants comprise nearly 20% of the Jewish population and 15% of the overall Israeli population. Since 2018, more than half of all immigrants to Israel (56%) have come from the FSU [11]. Previous studies [12] have reported that FSU families in Israel often face socioeconomic challenges that place them at high risk. For example, in the first years after immigration to Israel, new FSU immigrants often struggle financially and professionally, regardless of their educational or professional credentials. Many FSU immigrants also face discrimination based on the perceived economic and cultural threats that they pose [12]. Further, a recent study [5] suggests that recent immigrant families in Israel are more vulnerable to COVID-19 related risks (e.g., higher levels of COVID-19 distress) compared to native-born Israeli families. Taken together, it is important for prevention scientists to understand how various COVID-19 related factors impacts family functioning among recent immigrant families from the FSU in Israel.

The present study was grounded in an ecodevelopmental perspective. Ecodevelopmental models grew out of attempts to integrate family-systemic perspectives [13] with a more expansive ecological focus on identifying risk and protective factors across and within different contexts and their complex effects on individuals' developmental processes [14]. In the present study, we examine the various protective and risk effects of COVID-19 (within individual, family, and social contexts) on family functioning among a sample of recent immigrant families in Israel. Understanding COVID-19 related risk and protective factors across multiple contexts, and their effects on family functioning, has implications for the design of successful family-based interventions to reduce pandemic-related stress. We first conceptualize family functioning and then discuss how multiple risk and protective factors of COVID-19 impact family functioning among immigrant families in Israel.

## Family functioning

*Family functioning* has been framed as the confluence of parent-adolescent communication, parental involvement, positive parenting, and the absence of family conflict [15]. Communication involves the free exchange of thoughts, feelings, and ideas between youth and parents, as well as a belief that one does not need to restrain or censor oneself when communicating with family members [16]. Parental involvement entails parents asking youth what they did (or were planning to do) during the day, as well as displaying genuine interest in their thoughts and feelings [17]. Positive parenting involves providing rewards for desirable behavior and gently redirecting undesirable behavior [18]. Family conflict refers to a consistent pattern of arguing, fighting, or yelling among family members [19]. Research has found that family conflict relates negatively to family communication, parental involvement, and positive parenting among immigrant families [20].

Previous studies have found that parents' and adolescents' different roles and goals within the family often lead to divergent perceptions of family functioning. For example, parents might perceive their communication with their adolescent as open and warm, whereas the adolescent might not share this perception [21]. However, many studies have focused primarily on either adolescents' or parents' perception of family functioning, and it is essential to consider both perspectives within a single study. Further, less is known about parents' and adolescents' perceptions of family functioning during the COVID-19 pandemic.

The COVID-19 pandemic may place immigrant families in unique circumstances [22]. For example, a recent study [5] found that the COVID-19 pandemic has resulted in immigrant families receiving fewer social supports from community (such as mental health services) compared to native-born families, which may in turn be negatively related to their family functioning. To provide essential services to immigrant families in Israel and other countries, it is important to understand how the COVID-19 pandemic has affected parents' and adolescents' perception of family functioning within immigrant families.

## The role of ecodevelopmental context: Effects of COVID-19 risk and protective factors on family functioning

Ecodevelopmental theory emphasizes two types of contextual effects: (a) social–ecological effects and (b) social interactional effects. Social–ecological effects are based on Bronfenbrenner's social-contextual theory [23], which emphasizes the unique effects of multiple contexts (e.g., individual, family, and social contexts) influencing individual developmental processes [14]. Social interactional effects emphasize the interplay within and across multiple contexts, as well as their effects on developmental processes [24]. We examine both effects in the present

study. Here, we describe risk and protective COVID-19-related contextual factors vis-à-vis family functioning.

First, in terms of *social–ecological* effects, COVID-19 distress can be considered as a key *individual-context* COVID-19 risk factor. For example, many family members have reported widespread emotional distress in response to the COVID-19 pandemic [10]. Emotional distress (e.g., high levels of COVID-related anxiety) among family members may adversely affect family functioning (or vice versa) [25]. Importantly, given the interdependence of stress processes between parents and adolescents (e.g., parent-adolescent interdependence) [26], the associations between COVID-19 distress and family functioning may occur dynamically within the family context. For example, parents' distress may be disruptive not only to their own perceptions of family functioning, but also to adolescents' perceptions of family functioning, and vice versa.

Next, parents' employment can be considered as unique *family-context* COVID-19 risk and protective factors. For example, the COVID-19 pandemic has led to high rates of unemployment [27], which would likely directly compromise parents' perceptions of family functioning. That is, when parents are stressed or worried from financial strain, their attention to and patience for their children likely decreases. Because adolescents' perception of family functioning is impacted indirectly by parents' financial strains [28], it stands to reason that their perceptions of family functioning would be also impacted from family financial strain. Instead, parents' remote work arrangements (working from home) can be considered as *family-level* COVID-19 protective factor. Parents who kept their job and work at home may have more time to spend with their families. That is, when parents do not have to commute and can take breaks during the day to spend time with their youth, the parent-adolescent relationship may potentially improve. Finally, COVID-19 prevalence (new cases and deaths) represents a *social-context* COVID-19 related risk factor. Given the widespread media coverage surrounding COVID-19, and the anxiety resulting from consuming this coverage [29], it stands to reason that weeks with greater numbers of new COVID-19 cases and deaths would be associated with poorer family functioning.

In terms of *social interactional* effects, previous literature suggests not only that negative social contextual factors would exert independent effects influencing individuals' outcomes, but also that negative social contextual factors at different levels might moderate each other's effects [30]. During the COVID-19 pandemic, negative social contexts such as COVID-19 prevalence (deaths and new cases) may modify negative or positive effects of other COVID-19 related factors on family functioning. For example, on days with greater numbers of new COVID-19 cases and deaths, the negative effects of individual- (e.g., COVID-19 distress) and family-contexts (e.g., parental job loss) COVID stressors on family functioning might be amplified. When more COVID-19 cases and deaths are reported, the effects of COVID-19 distress may be amplified, and the positive effects of parents' remote work arrangements (working from home) on family functioning may be reduced.

## The present study: Research aims

The present study was guided by two aims. The first aim was to examine unique effects (social-ecological effects) of COVID-related risk and protective factors (COVID-19 distress, parent's employment, and COVID-19 prevalence) on parent and adolescent reports of family functioning among recently arrived post-Soviet immigrant families in Israel during the summer of 2020 (June to August). The second aim was to examine the moderating effect (social interactional effects) of COVID-19 prevalence in the negative and positive effects of COVID-related individual- and family-level factors on family functioning.

## Method

### Study design and participants

Families were recruited via social media, word-of-mouth, and referrals during June, July, and August of 2020 throughout Israel. To be eligible for the study, families must (a) have one parent and one adolescent between the ages of 12–15 willing to participate, (b) have migrated from a former Soviet country (e.g., Russia, Ukraine) to Israel during the 5 years prior to the assessment, and (c) be able to speak/write in Russian or Hebrew. Questionnaires were sent to families after a phone call with a research team member to clarify that the family met study enrollment criteria. Parents were paid $23, and children $10, as gratitude for their participation. The final sample providing information on study variables included in the current study consisted of 160 post-Soviet immigrant families (parents and adolescents) in Israel (see participants' demographic information in Table 1). The study was approved by the research ethics committee of the Hebrew University of Jerusalem (Jerusalem, Israel).

### Measures

**Family functioning.** Family functioning was assessed using adolescent and parent reports of four indicators, including parent-adolescent communication, parental involvement, positive parenting, and family conflict. The same items were presented to both parents and adolescents, with rewording to reflect each person's role in the dyad. For example, the item "how often do you talk to your child?" would be presented to youth as "how often does your parent talk to you?".

Parent–adolescent communication (hereafter referred as to P-A communication) (10 items; $\alpha$s = .92 and .88 for parents and adolescents, respectively) was assessed using the Parent–Adolescent Communication Scale [16]. Sample items include "I can discuss my beliefs with my child without feeling restrained or embarrassed." Response choices ranged from 1 (*Strongly Disagree*) to 5 (*Strongly Agree*). Parental involvement (11 items; $\alpha$s = .84 and .83 for parents and adolescents, respectively) and positive parenting (6 items; $\alpha$s = .73 for both parents and adolescents) were assessed using the corresponding subscales from the Parenting Practices Scale [17]. The parental involvement subscale includes items for the parent such as "How often do you and your child do things together at home?", with a response scale ranging from 1 (Never) to 5 (Always). The positive parenting subscale measures rewarding and acknowledging positive adolescent behaviors. Sample items for the parent include "When your child has done something that you like or approve of, do you say something nice about it, praise, or give approval?" Response choices ranged from 1 (Never) to 5 (Always). Family conflict (7 items; $\alpha$s = .73 and .77 for parents and adolescents, respectively) was assessed using the Conflict subscale of the Family Environment Scale (FES) [19]. A high score indicates more openly expressed anger, aggression, and conflict (e.g., "We fight a lot in our family"). Parents and adolescents completed the same set of items using a 5-point response scale from 1 (Strongly disagree) to 5 (Strongly agree).

**COVID-19 distress.** To assess parents' and adolescents' pandemic-related emotional distress, we used the three dichotomously scored items from the pandemic stress index (PSI) [31]. Both parents and adolescents were asked to report on depressive symptoms, anxiety, and insufficient sleep duration during COVID-19 (No = 0; Yes = 1). To obtain total scores, items are summed across the three items. Reliabilities ($\omega$) for COVID-19 distress were .80 and .70 for adolescents and parents, respectively.

**COVID-19 employment.** We asked parents about their employment during COVID-19. Two items were used: (a) How many adults who live with you have stopped working because of the coronavirus? and (b) How many adults who live with you work from home? We

**Table 1. Participants demographic information (*n* = 160).**

| Variables | Means (SD) or % | Min. / Max. |
|---|---|---|
| Parents | | |
| Female | 87.5% | |
| Age | 41.75 (5.25) | 30 / 50 |
| Relationship with Child (Not respond: 3.2%) | | |
| Mother | 85.6% | |
| Father | 10.6% | |
| Stepfather | .6% | |
| Education level | | |
| High school (including not completed) | 61.0% | |
| College (including graduate college) | 39.0% | |
| Marital Status (Not respond: 2.3%) | | |
| Married | 76.3% | |
| Divorced /Living separately / Never married | 21.4% | |
| Adolescents | | |
| Female | 49.4% | |
| Age | 13.61 (1.25) | 11 / 17 |
| Median grade | 8th grade | 6th grade / 11th grade |
| Household Information | | |
| Country of Origin | | |
| Russian | 60.6% | |
| Ukrainian | 34.4% | |
| Belarusian | 4.4% | |
| Years of immigration to Israel | 2.63 (1.48) | 0.00 / 5.00 |
| Family (monthly) Income (unit: Shekel [≈$.30]; Not respond: 2.5%) | | |
| Less than 10,000 | 48.1% | |
| 10,000 to 15,000 | 37.5% | |
| 15,000 to 20,000 | 10.0% | |
| Over 20,000 | 1.9% | |
| Numbers of children in house (Median) | 2 | 1 / 4 |
| People to room ratio [a] | 1.46 | |
| Household COVID-19 Information | | |
| Employment status affected by COVID-19 | | |
| Stopped working | 32.3% | |
| Working from home | 30.4% | |
| Any family members died due to COVID-19 | 0.7% | |
| Any family members had been diagnosed with COVID-19 (Positive) | 0.0% | |
| Time social distancing (Not respond: 5.0% and 5.6% for parents and adolescent, respectively) | | |
| No (Parent, Adolescent) | 64.0%, 45.0% | |
| One month or less (Parent, Adolescent) | 3.1%, 10.6% | |
| More than one month (Parent, Adolescent) | 26.9%, 38.8% | |
| COVID-19-related-distress (Parent, Adolescent) | .62 (.83), .59 (.74) | .00 / 3.00, .00 / 3.00 |
| COVID-19 Daily Prevalence [b] | | |
| New cases | 1187.92 (592.97) | 157 / 21000 |
| Death | 4.74 (4.26) | 0.00 / 14.00 |
| Family Functioning | | |
| Parent-Adolescent (P-A) Communication (Parent, Adolescent) | 3.94 (.58), 3.70 (.76) | 2.30 / 5.00, 1.50 / 5.00 |
| Parental Involvement (Parent, Adolescent) | 4.27 (.45), 3.89 (.64) | 2.82 / 5.00, 1.70 / 5.00 |

(*Continued*)

**Table 1.** (Continued)

| Variables | Means (SD) or % | Min. / Max. |
|---|---|---|
| Positive parenting (Parent, Adolescent) | 4.24 (.51), 3.77 (.73) | 2.33 / 5.00, 1.50 / 5.00 |
| Family conflict (Parent, Adolescent) | 2.24 (.50), 2.56 (.52) | 1.22 / 5.00, 1.00 / 5.00 |

[a] Numbers of people living in home divided by numbers of bedrooms in home.
[b] Jun. 2020 to Aug. 2020.

dichotomized both of these items (None = 0; One or more [unemployment; working from home] = 1).

**COVID-19 daily prevalence.** Information about number of daily new cases and number daily new deaths of COVID-19 during the week of assessment was extracted from the official Israeli Ministry of Health site (https://datadashboard.health.gov.il/COVID-19/general?utm_source=go.gov.il&utm_medium=referral).

## Analysis plan

Our analytic plan proceeded in three steps. We first examined means (or proportions) and standard deviations for four family functioning indicators (P-A communication, parental involvement, positive parenting, and family conflict) for both parents and adolescents. We also computed correlations among the key study variables (COVID-19 distress, employment, daily prevalence, and family functioning indicators). Second, we examined the associations between participant demographic characteristics and these four family functioning indicators by performing a series of bivariate analyses such as ANOVAs (for categorical variables) and Pearson correlations (for continuous variables) analyses. If there were any significant associations between demographic characteristics and family functioning indicators, we entered those demographic variables into the subsequent analyses to control for the effects of significant demographic variables.

Third, to examine unique effects of multiple contextual COVID-19 factors on four family functioning indicators, four multiple regression analyses were performed. In each regression model, four COVID-19 related factors were specified as independent variables: (a) COVID-19 distress, (b) stopped working, (c) working from home, and (d) national COVID-19 prevalence during the week when the family was assessed. Also, in each model, one of the four family functioning variables was specified as the dependent variable: (a) P-A communication, (b) parental involvement, (c) positive parenting, and (d) family conflict. To account for parent-adolescent interdependence of COVID-19 distress and family functioning processes (e.g., parent's COVID-19 distress → adolescents' perception of family functioning; adolescent's COVID-19 distress → parents' perception of family functioning), we estimated two separate regression models for each family functioning indicator by specifying parents' and adolescents' family functioning variables separately in the regression models. In each regression model, both parents' and adolescents' COVID-19 distress scores were entered simultaneously as independent variables.

Also, we examined the moderating effects of COVID-19 prevalence (i.e., new cases and deaths) on the associations between individual and family-level COVID-19 related factors and family functioning by adding a series of interaction terms to each of the regression models. In the moderating effect models, COVID-19 related factors were centered prior to computing the interaction terms, and each interaction was examined in a separate analysis to prevent multi-collinearity problems [32]. To interpret significant moderated effects, simple slopes for

COVID-19 family risks were computed at high and low levels of COVID-19 prevalence (using the mean split) [32]. Wald $\chi^2$ tests were used to examine the significant difference in simple slopes. Overall, with 80% power and $\alpha$ = .05, our sample size ($n$ = 160) was able to detect a significant interaction regression coefficient of |0.20|, which represents a small effect [33]. The amount of missing data for the family functioning indicators was small (on average 5% across family functioning indicators). Little's MCAR chi-square test [34] yielded a nonsignificant result ($\chi^2$(27) = 34.128, $p$ = .16), indicating no statistically reliable missing patterns for family functioning. All analyses were therefore conducted in Mplus (version 8.00) [35] utilizing the full-information-maximum-likelihood (FIML) method. All models were saturated for both parents and adolescents. Standardized coefficients ($\beta$) were reported as effect sizes.

## Results

### Preliminary analysis

Table 1 displays means for our four family functioning variables: (a) P-A communication, (b) parental involvement, (c) positive parenting, and (d) family conflict. Several significant correlations were found between COVID-related factors and family functioning indicators (S1 Table). Additionally, we found positive correlations in corresponding family functioning indicators between parents and adolescents (e.g., P-A communication reports between parents and adolescents) (S1 Table). The correlation between parent and adolescent COVID-19 distress scores was .17, $p$ < .05. S2 Table displays the mean differences (or correlations) of family functioning indicators with participants' sociodemographic variables. In general, for both parents and adolescents, females reported more positive (or less negative) family functioning compared to males. For example, mothers reported higher levels of parental involvement ($t$ = 3.49, $p$ < .001, $d$ = .82) and positive parenting ($t$ = 2.75, $p$ < .001, $d$ = .59) compared to fathers. In addition, girls reported lower levels of family conflict compared to boys ($t$ = 10.85, $p$ < .001; $d$ = .53). No other demographic variables were significantly related to family functioning.

### Unique effects of COVID-19 related factors

The results indicated some significant associations between COVID-19 factors and family functioning indicators. In general, these associations were more widespread for adolescents than for parents (see corresponding coefficients in Tables 2 and 3). For example, parent reports of COVID-19 distress were positively associated with their own reports of family conflict ($\beta$ = .18, $p$ < .05). Instead, adolescent reports of COVID-19 distress were negatively associated with their own reports of parental involvement ($\beta$ = -.17, $p$ < .05) and positive parenting ($\beta$ = -.19, $p$ < .05). In addition, adolescents' COVID-19 distress was positively associated with their own reports of family conflict ($\beta$ = .25, $p$ < .01). In addition, we found one parent-adolescent interdependence association between COVID-19 distress and family functioning. That is, adolescent reports of COVID-19 distress were negatively associated with parent reports of parental involvement ($\beta$ = -.18, $p$ < .05). In addition, we found that parents' employment status and remote work arrangements were only associated with their own reports of family functioning. For example, having stopped working was positively associated with parents' own reports of family conflicts ($\beta$ = .16, $p$ < .05). Conversely, working from home was positively associated with parent reports of P-A communication ($\beta$ = .17, $p$ < .05).

Next, results indicated significant associations between COVID-19 prevalence and family functioning indicators (see Tables 2 and 3). However, patterns of associations were different depending on type of COVID-19 prevalence (new cases versus deaths) and respondent (parent versus adolescent). For example, results indicated that number of new COVID-19 cases during the week of assessment were significantly associated with all four adolescent-reported family

functioning indicators: P-A communication ($\beta$ = -.26, $p < .05$), parental involvement ($\beta$ = -.31, $p < .05$), positive parenting ($\beta$ = -.29, $p < .05$), and family conflict ($\beta$ = .30, $p < .05$). Conversely, new cases of COVID-19 were negatively associated only with parent reports of P-A communication ($\beta$ = -.22, $p < .05$). In addition, results indicated that deaths due to COVID-19 during the week of assessment were negatively associated with adolescent reports of family conflict ($\beta$ = -.31, $p < .01$).

## Moderating effects of COVID-19 prevalence

Tables 2 and 3 present the moderating effects of COVID-19 prevalence (new cases and deaths during the week of assessment) on the associations between COVID-19 related factors

**Table 2. Unique and moderating effects of COVID-19 related factors on parent's perception of family functioning.**

| | P-A communication *(Parent report)* | | Parental Involvement *(Parent report)* | | Positive parenting *(Parent report)* | | Family conflict *(Parent report)* | |
|---|---|---|---|---|---|---|---|---|
| Predictors | β | 95% CI | β | 95% CI | β | 95% CI | β | 95% CI |
| **Unique Effects of COVID 19 related Factors** | | | | | | | | |
| **COVID-19 Distress** | | | | | | | | |
| COVID-19 distress (Parent report) | -.07 | -.23, .10 | -.03 | -.18, .13 | -.11 | -.27, .04 | **.18**\* | .04, .33 |
| COVID-19 distress (Adolescent report) | -.11 | -.25, .04 | **-.18**\* | -.31, -.04 | -.09 | -.28, .09 | -.07 | -.21, .08 |
| **COVID-19 Employment** | | | | | | | | |
| Stopped working | .04 | -.11, .19 | .003 | -.15, .16 | -.08 | -.25, .09 | **.16**\* | .01, .32 |
| Working from home | **.17**\* | .01, .33 | .05 | -.08, .19 | .04 | -.11, .20 | -.07 | -.23, .09 |
| **COVID-19 Daily Prevalence** | | | | | | | | |
| New cases | **-.22**\* | -.43, -.01 | -.08 | -.27, .11 | -.07 | -.29, .16 | .15 | -.06, .37 |
| Death | .09 | -.15, .33 | .06 | -.13, .25 | .01 | -.22, .23 | -.09 | -.33, .13 |
| **Controls (Parent Gender)** | | | | | | | | |
| Female (vs. Male) | .11 | -.05, .27 | **.30**\*\*\* | .12, .49 | **.26**\*\*\* | .05, .46 | -.06 | -.22, .10 |
| **Moderating Effects of COVID-19 Prevalence** | | | | | | | | |
| **COVID-19 Distress × Prevalence** | | | | | | | | |
| Parent Distress | | | | | | | | |
| COVID-19 Distress × New cases | .07 | -.08, .23 | .05 | -.09, .18 | .07 | -.09, .23 | -.10 | -.25, .04 |
| COVID-19 Distress × Deaths | .03 | -.14, .20 | -.07 | -.21, .07 | -.05 | -.22, .12 | -.07 | -.23, .10 |
| Adolescent Distress | | | | | | | | |
| COVID-19 Distress × New cases | -.01 | -.15, .13 | -.07 | -.20, .06 | .10 | -.09, .28 | -.06 | -.22, .11 |
| COVID-19 Distress × Deaths | .08 | -.02, .05 | .003 | -.02, .02 | .06 | -.02, .04 | -.13 | -.05, .01 |
| **COVID-19 Employment Status × Daily Prevalence** | | | | | | | | |
| Stopped working | | | | | | | | |
| Stopped working × New cases | -.03 | -.18, .13 | -.09 | -.24, .05 | -.15 | -.30, .01 | -.03 | -.22, .16 |
| Stopped working × Deaths | -.05 | -.21, .11 | -.07 | -.22, .08 | -.15 | -.33. 01 | -.03 | -.24, .19 |
| Working from home | | | | | | | | |
| Working from home × New cases | -.15† | -.32, .01 | -.03 | -.16, .11 | -.09 | -.26, .07 | .13 | -.04, .30 |
| Working from home × Deaths | **-.18**\* | -.37, -.01 | **-.17**\* | -.30, -.03 | **-.22**\*\* | -.38, -.06 | .15 | -.03, .34 |
| **R-square** | .15 | | .19 | | .19 | | .12 | |

$\beta$ = Standardized regression coefficients. Bold values represent statistically significant coefficients. Grayed values represent no statistical significance.

†$p < .10$.

\*$p < .05$.

\*\*$p < .01$.

\*\*\*$p < .001$.

**Table 3. Unique and moderating effects of COVID-19 related factors on adolescent's perception of family functioning.**

| Predictors | P-A communication (Adolescent report) | | Parental Involvement (Adolescent report) | | Positive parenting (Adolescent report) | | Family conflict (Adolescent report) | |
|---|---|---|---|---|---|---|---|---|
| | β | 95% CI | β | 95% CI | β | 95% CI | β | 95% CI |
| **Unique Effects of COVID 19 related Factors** | | | | | | | | |
| **COVID-19 Distress** | | | | | | | | |
| COVID-19 Distress (Parent report) | .07 | -.10, .25 | .05 | -.11, .21 | -.05 | -.20, .11 | .04 | -.13, .21 |
| COVID-19 Distress (Adolescent report) | -.12 | -.29. .06 | **-.17**\* | -.32, -.02 | **-.19**\* | -.36, -.02 | **.25**\*\* | .08, .42 |
| **COVID-19 Employment** | | | | | | | | |
| Stopped working | .04 | -.12, .21 | .04 | -.13, .20 | -.05 | -.22, .11 | .01 | -.16, .17 |
| Working from home | -.16 | -.35, .03 | -.08 | -.27, .10 | -.14 | -.30, .03 | .11 | -.07, .28 |
| **COVID-19 Daily Prevalence** | | | | | | | | |
| New cases | **-.26**\* | -.53, -.01 | **-.31**\* | -.56, -.05 | **-.29**\* | -.55, -.03 | **.30**\* | .06, .55 |
| Death | .21 | -.06, .48 | .22 | -.03, .48 | .20 | -.07, .47 | **-.31**\*\* | -.57, -.05 |
| **Controls (Adolescent Gender)** | | | | | | | | |
| Female (vs. Male) | -.05 | -.21, .12 | .06 | -.11, .23 | -.05 | -.22, .12 | **-.25**\*\*\* | -.41, -.10 |
| **Moderating Effects of COVID-19 Prevalence** | | | | | | | | |
| **COVID-19 Distress × Prevalence** | | | | | | | | |
| Parent Distress | | | | | | | | |
| COVID-19 Distress × New cases | .07 | -.09, .23 | .01 | -.12, .14 | -.13 | -.28, .01 | -.07 | -.21, .07 |
| COVID-19 Distress × Deaths | .02 | -.16, .19 | -.13 | -.28, .03 | **-.19**\* | -.34, -.04 | -.01 | -.17, .16 |
| Adolescent Distress | | | | | | | | |
| COVID-19 Distress × New cases | **.19**\*\* | .05, .34 | .09 | -.06, .24 | .09 | -.10, .28 | -.04 | -.21, .13 |
| COVID-19 Distress × Deaths | .09 | -.04, .24 | -.04 | -.17, .09 | .01 | -.19, .22 | -.04 | -.17, .09 |
| **COVID-19 Employment Status × Daily Prevalence** | | | | | | | | |
| Stopped working | | | | | | | | |
| Stopped working × New cases | -.05 | -.22, .10 | -.03 | -.19, .13 | -.001 | -.15, .15 | -.02 | -.16, .14 |
| Stopped working × Deaths | -.11 | -.26, .03 | -.03 | -.19, .13 | -.08 | -.24, .08 | .04 | -.12, .20 |
| Working from home | | | | | | | | |
| Working from home × New cases | -.05 | -.22, .14 | .004 | -.15, .16 | -.07 | -.23, .08 | .09 | -.07, .25 |
| Working from home × Deaths | -.10 | -.28, .07 | **-.15**\* | -.30, -.01 | **-.18**\* | -.34, -.03 | **.24**\*\* | .09, .39 |
| **R-square** | .14 | | .17 | | .19 | | .24 | |

*β* = Standardized regression coefficients. Bold values represent statistically significant coefficients. Grayed values represent no statistical significance.

†$p < .10$.

\*$p < .05$.

\*\*$p < .01$.

\*\*\*$p < .001$.

(COVID-19 distress and employment) and family functioning. In terms of COVID-19 distress, new COVID-19 cases positively moderated the associations between adolescent reports of COVID-19 distress and their own reports of P-A communication ($\beta = .19$, $p < .01$; see Table 3). This moderating effect is plotted in panel *a* of Fig 1. Adolescents' COVID-19 distress was negatively associated with adolescent-reported P-A communication when new cases were below average (Wald $\chi^2 = 10.14$, $p < .001$). However, this negative association did not emerge when new COVID-19 cases were above average (Wald $\chi^2 = 1.04$, $p = .31$). Further, deaths from COVID-19 negatively moderated the associations between parent reports of COVID-19 distress and adolescent reports of positive parenting ($\beta = -.19$, $p < .01$; see Table 3). This moderating effect is plotted in panel *b* of Fig 1. Parents' COVID-19 distress was not associated with adolescent reports of positive parenting when COVID-19 deaths were below average (Wald

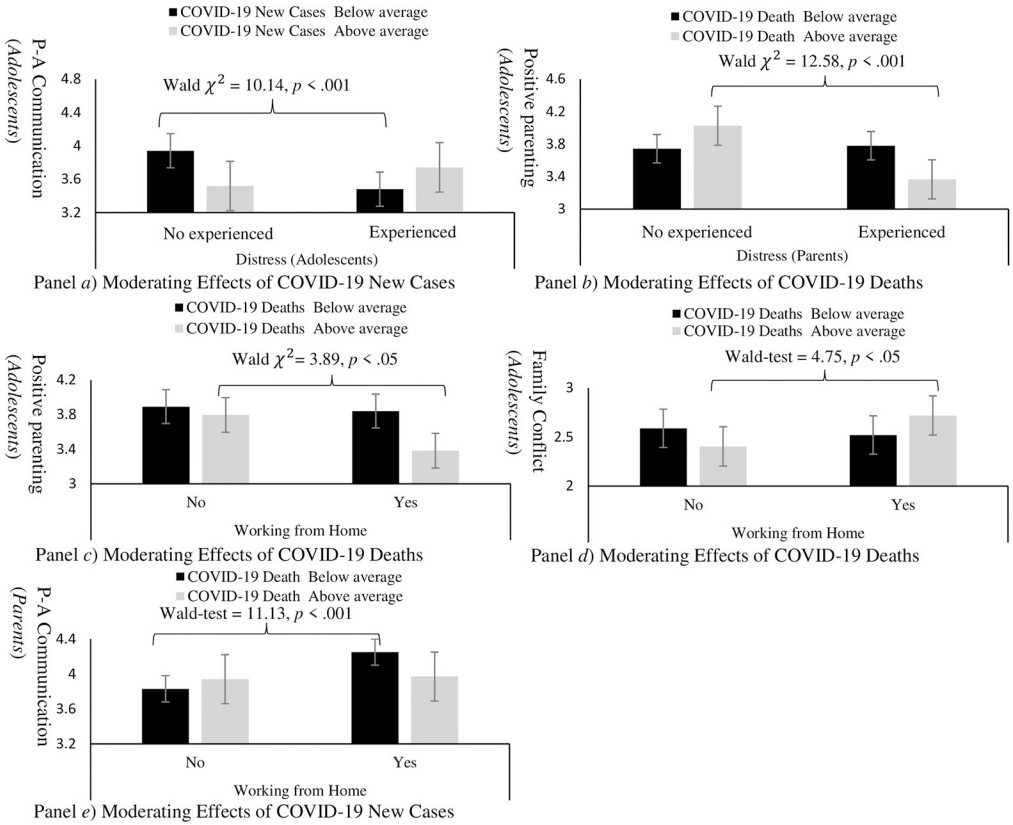

Note. Error bars represent 95% Confidence intervals. For plotting simple slopes, COVID-19 distress was dichotomized (No experienced [coded as 0] vs. Experience of any symptoms of depressive symptoms, anxiety, or insufficient sleep [coded as 1]).

**Fig 1. Moderating effects of COVID-19 daily prevalence.**

$\chi^2 = .64, p = .80$). However, when COVID-19 deaths were above average, parents' COVID-19 distress was negatively associated with adolescents' reports of positive parenting (Wald $\chi^2 = 12.58, p < .001$).

Next, results indicated that COVID-19 deaths negatively moderated effects of working from home on both parents' and adolescents' reports of positive parenting and parental involvement (see corresponding four coefficients in Tables 2 and 3). As an example, moderating effects of COVID-19 deaths on the associations between working from home and adolescent reports of positive parenting are plotted in panel *c* of Fig 1 ($\beta = -.18, p < .01$; see Table 3). Working from home was negatively associated with adolescent-reported positive parenting when COVID-19 deaths were above average (Wald $\chi^2 = 3.89, p < .05$). However, this negative association did not emerge when COVID-19 deaths were below average (Wald $\chi^2 = .08, p = .78$). Shapes of negative moderating effects of COVID-19 death on adolescents' reports of parental involvement ($\beta = -.15, p < .05$), parents' reports of parental involvement ($\beta = -.17, p < .05$), and parents' reports of positive parenting ($\beta = -.22, p < .01$) were similar (S1 Fig).

Results also indicated that COVID-19 deaths positively moderated the associations between working from home and adolescent-reported family conflict ($\beta = .24, p < .01$; see Table 3). This moderating effect is plotted in panel *d* of Fig 1. Working from home were positively associated with adolescent-reported family conflict when COVID-19 deaths were above average (Wald $\chi^2 = 4.75, p < .05$). However, this positive association did not emerge when COVID-19

deaths were below average (Wald $\chi^2$ = 0.29, $p$ = .58). In addition, COVID-19 deaths negatively moderated the associations between working from home and parent reports of P-A communication ($\beta$ = -.18, $p$ < .05; see Table 2). This negative moderating effect indicates that working from home were positively associated with parent-reported P-A communication when deaths were below average (Wald $\chi^2$ = 11.13, $p$ < .001; see panel $e$ of Fig 1). However, this positive association did not emerge when deaths were above average (Wald $\chi^2$ = 0.03, $p$ = .85).

## Gender differences in family functioning

In general, gender differences in family functioning remained significant even after controlling for unique and moderating effects of COVID-19 factors. That is, mothers reported higher levels of parental involvement ($\beta$ = .30, $p$ < .001) and positive parenting ($\beta$ = .26, $p$ < .001) compared to fathers (see Table 2). In addition, girls reported lower levels of family conflict compared to boys ($\beta$ = -.25, $p$ < .001). Other variables were not significantly related to family functioning (see Table 3).

## Discussion

With a sample of post-Soviet immigrant families in Israel and grounded in an ecodevelopment perspective, we aimed to examine the effects of COVID-19 related risk and protective factors on family functioning. To pursue these research aims, we examined unique (social-ecological) and moderating (social interactional) effects of social contexts (COVID-19 prevalence and deaths) on effects of individual- (COVID-19 distress) and family- (employment) context factors on family functioning in both parents and adolescents.

### COVID-19 distress and family functioning

We found that COVID-19 distress was associated with both parents' and adolescents' perceptions of family functioning. Results suggested both similarities and dissimilarities between parent and adolescent reports vis-à-vis distress-family functioning associations. Regarding similarities in the distress-family functioning processes, results indicated positive associations between COVID-19 distress and family conflicts for both parents and adolescents. That is, parents and adolescents who worried more about COVID-19 were more likely to report family conflicts. Uncertainty due to the COVID-19 pandemic may affect both parents' and adolescents' perceptions of family conflicts [25]. However, there were also dissimilarities across reporters in distress-family functioning associations. For example, adolescents' reports of COVID-19 distress were negatively associated with their own reports of positive parenting and parental involvement. These negative associations did not emerge for parent reports. These results suggest that links between COVID-19 distress and family functioning were more prevalent for adolescents than for parents.

In addition, our findings suggest parent-adolescent interdependence (between-reporter transactional associations) [26] between COVID-19 distress and family functioning. Specifically, as adolescents reported higher levels of COVID-19 distress, their parents reported lower levels of their involvement with them. This result is consistent with previous findings. For example, Pineda et al. [36] suggested that parents of distressed adolescents were generally less positive with their youth and reduced their involvement in the adolescent's life. However, these studies primarily used adolescents' reports of their parents' behaviors. The current findings extend our understanding by identifying that adolescents' distress was not only associated with their own perception of parental involvement, but also may be linked to parents' own perception of their involvement. Family systems perspectives posit that the family is a "complex, integrated whole", wherein individual family members are interdependent and exert a

continuous and reciprocal influence on one another [13]. Our findings for adolescents' COVID-related distress and parental involvement appear consistent with a family systems perspective.

## Employment and family functioning

We also found that parental unemployment was positively associated with parents' perception of family conflicts. This result is consistent with family stress theory, which emphasizes negative effects of family economic adversity, such as parent job loss or unemployment, on family relationships [28]. Interestingly, our results did not indicate any associations between unemployment and positive indicators of family functioning from either parents' or adolescents' perspectives. This may be due to financial support from the Israeli government and from the private sector, where some companies created their own compensation funds to cover employees' salaries during the pandemic [37]. These supports may have offset or delayed the effects of family financial strain on family functioning.

Moreover, we found positive (protective) effects of remote work arrangements on family functioning. That is, while working from home, parents reported more communication with their adolescents. We did not find significant associations between working from home and other positive family functioning indicators (parental involvement and positive parenting). However, given the positive associations among P-A communication, parental involvement, and positive parenting, working from home may be positively associated with parental involvement and positive parenting indirectly through P-A communication. Longitudinal studies are needed to examine how adult employment (unemployment and remote work arrangements) affected by COVID-19 influence changes in family functioning. Taken together, the present results suggest that work-related arrangements may be more likely to be linked with parents' perception of family functioning.

## COVID-19 prevalence and family functioning

Our results indicate inverse associations between the number of new COVID-19 cases during the week of assessment and family functioning in both parents and adolescents. However, these associations differ across family functioning indicators and between parents and adolescents. For example, on weeks when more COVID-19 cases were reported, parents indicated less communication with adolescents. On the other hand, on weeks when more COVID-19 cases were reported, adolescents not only indicated less communication with their parents, but also indicated that their parents engaged less in parental involvement and in positive parenting practices. Adolescents also perceived more family conflict during weeks when more COVID-19 cases were reported. Similar patterns of distress-family functioning processes, these results suggest that effects of COVID-19 new cases on family functioning were more pervasive for adolescents than for parents. Taken together, adolescents' perception of family functioning appears to have been more strongly influenced by social contexts (COVID-19 new cases) compared to parents' perceptions.

In addition, we found that type of COVID-19 prevalence (new cases versus deaths) influenced family conflict differently, and that these patterns differed between parent and adolescent reports of conflict. Specifically, adolescent reported *more* family conflict when more COVID-19 new cases were reported. However, they reported *less* family conflict when more COVID-19 deaths were reported. Family members may have higher levels of fear when more COVID-19 deaths were reported [38] compared to when more COVID-19 cases were reported. To manage their fears or threats, family members may seek to foster a sense of unity and reduce conflicts with other family members. Yet, family (conflict) patterns before threat of

COVID-related death might resurface after the threat passes, suggesting the potential dynamic changes in family conflict over time. Interestingly, no associations were detected between COVID-19 prevalence (new cases and deaths) and parent-reported family conflict. Immigrant parents from the former Soviet Union may exhibit higher resilience due to experienced hardships in their country of origin [5]. Therefore, parents' behavior and their perceptions of family functioning may be less influenced by COVID-19 prevalence compared to adolescents' behaviors and perceptions.

## Moderating effects of COVID-19 prevalence on family functioning

In general, results indicate that negative social-context effects of the pandemic (such as COVID-19 deaths) modified the effects of remote work arrangements on family functioning. In terms of specific family functioning indicators, during weeks when more COVID-19 deaths were reported, positive effects of working from home on parent-reported P-A communication, parental involvement, and positive parenting were reduced. Similar effects emerged for adolescents' reports of parental involvement and positive parenting. Additionally, when more COVID-19 deaths were reported, working from home was positively associated with adolescent reports' family conflict. Taken together, these results suggest that when more COVID-19 deaths were reported, parents and adolescents may both be more concerned about their and their families' safety [38]–and the effects of more time spent together appear to become less beneficial.

Additionally, we found a positive moderating effect of new COVID-19 cases on the association between adolescents' COVID-19 distress and their own reports of P-A communication. During weeks when more COVID-19 new cases were reported, adolescents reported low levels of P-A communication regardless of their COVID-19 distress. However, during weeks when fewer new COVID-19 cases were reported, patterns of adolescents' reports of P-A communication were different. That is, adolescents reported lower levels of P-A communication in the presence of higher levels of COVID-19 distress. This result suggests that adolescents' COVID-19 distress reduced their perception of P-A communication even when fewer new COVID-19 cases were reported.

We also found a significant moderating effect of COVID-19 deaths through parents-adolescent interdependence associations [26]. That is, when more COVID-19 deaths were reported, parents' distress was negatively associated with adolescents' perception of positive parenting. One plausible explanation is that when COVID-19 deaths were higher, parents have been less able to contain their distress, such that it may have been more likely to impact their parenting behaviors. However, when COVID-19 deaths were lower, parents may be more able to contain their stress and limit impacts on their parenting interactions.

Additionally, we found gender differences in family functioning. For example, mother reported higher levels of parenting practices (parental involvement and positive parenting) compared to father. This result is consistent with findings reported by Tavassolie et al. [39]. Further, results indicated that female adolescents reported higher levels of family conflict compared to male adolescents. Consistent with this result, previous findings reported that daughters are less avoidant regarding conflict, and conflicts are mainly on everyday issues in which parents are involved [40]. Taken together, these findings confirmed that gender differences in family functioning occur similarly even during the COVID-19 pandemic.

## Strengths and limitations

Our findings make several important contributions to the family literature. First, our results support the eco-developmental perspective for recent immigrant families in Israel by examining unique and moderating effects of multiple COVID-19 related factors on family

functioning. Second, our results contribute to the literature by supporting the parent-adolescent interdependence perspective vis-à-vis COVID-19 distress and family functioning. Our study and findings help prevention scientists to understand family dynamics during the COVID-19 pandemic, and may also provide guidance to family therapists in working with immigrant families during a pandemic. Third, our results illustrate how social contexts (such as COVID-19 prevalence) can modify effects of family- (employment) and individual-level contexts (COVID-19 distress). Fourth, our data were obtained from three different sources–official government records, parent reports, and youth reports. Use of data from different sources helps to reduce shared method variance associated with measuring all constructs from the same reporter.

However, several limitations must also be acknowledged. First, our study was cross-sectional. This design does not allow us to examine dynamic changes in the effects of COVID-19 related risk and protective factors on family functioning over time. Second, the study sample consisted of recent immigrant families in Israel. Therefore, findings might not be applicable immigrants in other countries or other types of migrants (e.g., refugees). Fourth, the sample size was modest and did not permit us to include all of the family functioning indicators (of both parents and adolescents) within a single structural equation model. Fifth, the study sample was comprised primarily of mothers (87.5%). It is thus important to remain cautious when interpreting the results. For future study, it is important to replicate the results with a sample including more fathers. Sixth, the current study did not collect native-born families in Israel. Therefore, we do not know how our findings might have differed among recent immigrant families, as compared to native-born families, in Israel. Seventh, stress about elderly parents living abroad, with limited possibilities for visitation, can serve as a stressful event for migrants and can affect their family life during global pandemics. In future studies, this possibility should be examined in detail. Eighth, although we sampled families throughout Israel, we collected the data through convenience sampling methods. We do not know how representative our sample may be vis- à-vis the population of migrant families in Israel.

## Conclusion and implications

The present data are unique in that they capture family functioning among immigrant families in Israel approximately 3 months into Israel's response to the pandemic (June to August 2020), when Israel was locked down and people were not allowed to engage in social activities outside their homes. Our findings may hold key implications for clinical interventions targeting the family environment for immigrant families during a pandemic. Results suggest the importance of supporting immigrant families in managing everyday routines, as well as fostering positive family dynamics, during an emergency. Mental health providers would be well advised to direct special attention toward the unique impacts of multiple pandemic-related risks at the individual, family, and social factors on family functioning among immigrant families. In addition, results suggest that moderating effects of social contexts (COVID-19 prevalence) on family- (employment) and individual-level (COVID-19 distress) effects on family functioning. Thus, policy makers and mental health professionals working to prepare for potential disease outbreaks should be aware that multiple pandemic-related distress processes (individual, family and social risks) may exert different types of effects on different family members.

Importantly, results indicate that the increased prevalence for adolescents of COVID-19 risks (i.e., COVID-19 distress and prevalence) and family functioning, suggest that family-based interventionists may need to consider an adolescent-focused approach when they design intervention or make treatment programs during a pandemic. Further, given that the COVID-19 pandemic has increased social isolation, providing easily accessible and safe online contact

for all family members to receive support (e.g., emotional supports from friends, or professional mental health care) is of great importance for fostering positive coping strategies during a global pandemic. Taken together, the present study offers greater insight to the dynamic protective and risk effects of COVID-19-related contextual factors (social-, family- and individual-) on family functioning among immigrant families in Israel.

## Supporting information

**S1 Table. Correlations among COVID-19 distress, employment status, daily prevalence, and family functioning indicators.**
(DOCX)

**S2 Table. Means or correlations of family functioning among participants' demographic characteristics.**
(DOCX)

**S1 Fig. Moderating effects of COVID-19 death on associations between parents' remote work arrangements and family functioning.**
(TIF)

## Author Contributions

**Conceptualization:** Tae Kyoung Lee, Maya Benish-Weisman, Seth J. Schwartz.

**Formal analysis:** Tae Kyoung Lee.

**Investigation:** Tae Kyoung Lee.

**Methodology:** Tae Kyoung Lee.

**Writing – original draft:** Tae Kyoung Lee, Seth J. Schwartz.

**Writing – review & editing:** Tae Kyoung Lee, Maya Benish-Weisman, Saskia R. Vos, Maria Fernanda Garcia, Maria C. Duque Marquez, Ivonne A. Calderón, Tatiana Konshina, Einat Elizarov, Seth J. Schwartz.

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
