## [Decision Letter · Decision Letter 0]

5 Sep 2022

PONE-D-22-04347Psychological distress, employment, and family functioning during the COVID-19 outbreak among recent immigrant families in Israel: Moderating roles of COVID-19 prevalencePLOS ONE

Dear Dr. Lee,

Thank you for submitting your manuscript to PLOS ONE. After careful consideration, we feel that it has merit but does not fully meet PLOS ONE’s publication criteria as it currently stands. Therefore, we invite you to submit a revised version of the manuscript that addresses the points raised during the review process.

 Your manuscript has been reviewed by two peer-reviewers and their reports are appended below.  The reviewers commented that the manuscript requires further discussion regarding the reasoning for studying migrant populations in particular, and how these groups may be differently affected than the native population. Furthermore, the reviewers comment that the discussion and limitations of the study are expanded, for example to clarify how representative the study is for the whole migrant population. The reviewers have recommended a number of citations as a part of their review. We would recommend that you thoroughly evaluated these requested references and determine whether the articles are relevant to the current study. You may feel free to disregard references with tangible relevance to the study reported in the manuscript. Could you please revise the manuscript to carefully address the concerns raised?

We look forward to receiving your revised manuscript.

Kind regards,

Maria Elisabeth Johanna Zalm, Ph.D

Editorial Office

PLOS ONE

Reviewers' comments:

Reviewer's Responses to Questions

**Comments to the Author**

1. Is the manuscript technically sound, and do the data support the conclusions?

Reviewer #1: Yes

Reviewer #2: No

2. Has the statistical analysis been performed appropriately and rigorously? 

Reviewer #1: Yes

Reviewer #2: Yes

3. Have the authors made all data underlying the findings in their manuscript fully available?

Reviewer #1: Yes

Reviewer #2: Yes

4. Is the manuscript presented in an intelligible fashion and written in standard English?

Reviewer #1: Yes

Reviewer #2: Yes

5. Review Comments to the Author

Reviewer #1: The manuscript is written fluently and covers u particularly interesting topic. However, it is believed that the introductory part should also be expanded with other research developed at that time i other countries to better understand the phenomenon of family dynamics. By way of example, some work carried out in Italy at that time and recently published, in them can be found a rich and appropriate international literature

Rania, N., Parisi, R., & Lagomarsino, F. (2022). Mothers and Workers in the Time of COVID-19: Negotiating Motherhood within Smart Working. Journal of Contemporary Ethnography 1-31 doi.org/10.1177/08912416221075833.

Rania, N., Coppola, I., Lagomarsino, F., & Parisi, R. (2022). Family well-being during the COVID19lockdown in Italy: Gender differences and solidarity networks of care. Child & Family Social Work ,27(1), 1-10. https://doi.org/10.1111/cfs.1286710.

The analyses are conducted with methodoligical riogre, however, the presence of a higher percentage of mothers should be taken into account when discussing the limitations of the work.

Reviewer #2: This contribution is well-written and thorough. However, the main focus – determining how the covid-19 pandemic is related to the functioning of migrant families is not sufficiently well motivated and empirically studied. The main points of criticism are stated below:

- The theoretical arguments that are presented in the paper are generic and applicable to any group/ general population. I miss specific arguments of why should we study the migrant population (beyond the number in the entire population) and how the reaction to the pandemic might differ for this group as compared to native populations.

- Following that, I would suggest including the native population as a comparison group, to disentangle the difference. I am not sure, however, whether the data permits it. Alternatively, one can focus on this one group only and use more migration-specific arguments and indicators to show what is distinct about migrants’ situation.

- The contribution lacks links to the migration literature, and especially – transnationalism. Migrants who have migrated 5 years prior to the study and are in their 40-ies, might still have close ties to the family members living in the country of origin, especially parents (or did the parent generation migrate along?). Stress about elderly parents living abroad, with limited possibility to go and visit can be a stressful event for migrants and can affect their family life during the time of the global pandemic. Does the data set contain any information that would allow disentangling of those mechanisms?

- The authors use convenience sample – how representative is it for the whole migrant population? Is there a way to say anything about that?

- The conclusions about designing possible interventions to deal support migrant families in the time of the pandemic seem also very generic and applicable to general population. Again, what is so unique about this group, that does not apply to native population?

6. PLOS authors have the option to publish the peer review history of their article (what does this mean?). If published, this will include your full peer review and any attached files.

Reviewer #1: No

Reviewer #2: No

---

## [Author Response · Author response to Decision Letter 0]

20 Oct 2022

Reviewer #1:

Comment 1: It is believed that the introductory part should also be expanded with other research developed at that time other countries to better understand the phenomenon of family dynamics. By way of example, some work carried out in Italy at that time and recently published, in them can be found a rich and appropriate international literature.

Rania, N., Parisi, R., & Lagomarsino, F. (2022). Mothers and Workers in the Time of COVID-19: Negotiating Motherhood within Smart Working. Journal of Contemporary Ethnography, 1-31. doi.org/10.1177/08912416221075833.

Rania, N., Coppola, I., Lagomarsino, F., & Parisi, R. (2022). Family well-being during the COVID19 lockdown in Italy: Gender differences and solidarity networks of care. Child & Family Social Work ,27(1), 1-10. https://doi.org/10.1111/cfs.1286710.

Response: To understand better the family dynamics during the COVID-19 pandemic, we have now added international literature that the reviewer suggested (see page 3): 

“Some studies have begun to examine the impacts of COVID-19-related risks on family functioning among families in the U.S. [2] and Europe [3, 4]. For example, a recent study indicated that Italian families reported high levels of stress and tension during the COVID-19 pandemic [4]. However, less is known about COVID-19 effects on families in other countries outside the U.S. and Europe.”

Comment 2: The analyses are conducted with methodological rigor, however, the presence of a higher percentage of mothers should be taken into account when discussing the limitations of the work.

Response: In the revised manuscript, the high percentage of mother participants was acknowledged in the limitation section (see page 24): 

“Fifth, the study sample was comprised primarily of mothers (87.5%). It is thus important to remain cautious when interpreting the results. For future study, it is important to replicate the results with a sample including more fathers.”

Reviewer #2:

Comment 3: The theoretical arguments that are presented in the paper are generic and applicable to any group/ general population. I miss specific arguments of why we should study the migrant population (beyond the number in the entire population) and how the reaction to the pandemic might differ for this group as compared to native populations. 

Response: We agree that the findings of the study may be applicable to general populations given that such populations also have COVID-19-related risks (e.g., family financial strains) which can lead to negative effects on family functioning (e.g., family communication) [1]. However, previous studies have suggested that migrant populations, compared to native populations, may be more stressed due to additional pressures such as cultural stress (perceived discrimination and/or acculturative stress) in the context of COVID-19 [2], and that these pressures may exert more negative effects on family functioning. Importantly, less is known about effects of COVID-related risks among recent immigrant families. More importantly, as we mention in the manuscript, most recent studies have examined the impacts of COVID-19-related risks on family functioning among families in Westernized counties (such as the U.S. [2] and Europe [3, 4]). Therefore, to improve external validity (generalizability), we need to understand the effects of COVID-related risks on family functioning among immigrant populations outside of westernized countries. Accordingly, we examined effects of various COVID-19 stressors on family functioning among recent immigrant families in Israel. We have briefly summarized this in the introduction (see pages 3 and 4). 

“Some studies have begun to examine the impacts of COVID-19-related risks on family functioning among families in the U.S. [2] and Europe [3, 4]. For example, a recent study indicated that Italian families reported high levels of stress and tension during the COVID-19 pandemic [4]. However, less is known about COVID-19 effects on families in other countries outside the U.S. and Europe.”

“Further, a recent study [5] suggests that recent immigrant families in Israel are more vulnerable to COVID-19 related risks (e.g., higher levels of COVID-19 distress) compared to native-born Israeli families. Taken together, it is important for prevention scientists to understand how various COVID-19 related factors impacts family functioning among recent immigrant families from the FSU in Israel.”

References

1. Hussong AM, Midgette AJ, Richards AN, Petrie RC, Coffman JL, Thomas TE. COVID-19 Life Events Spill-Over on Family Functioning and Adolescent Adjustment. J Early Adolesc. 2022;42(3):359-388. doi:10.1177/02724316211036744

2. Kim Y, Lee H, Lee M. Social Support for Acculturative Stress, Job Stress, and Perceived Discrimination Among Migrant Workers Moderates COVID-19 Pandemic Depression. Int J Public Health. 2022;67:1604643. Published 2022 Aug 5. doi:10.3389/ijph.2022.1604643

Comment 4: Following that, I would suggest including the native population as a comparison group, to disentangle the difference. I am not sure, however, whether the data permits it. Alternatively, one can focus on this one group only and use more migration-specific arguments and indicators to show what is distinct about migrants’ situation.

Response: Unfortunately, a host national sample was not included when the data was collected. We now mention this in the limitation section (see page 24). 

“Sixth, the current study did not collect native-born families in Israel. Therefore, we do not know how our findings might have differed among recent immigrant families, as compared to native-born families, in Israel.”

Comment 5: The contribution lacks links to the migration literature, and especially – transnationalism. Migrants who have migrated 5 years prior to the study and are in their 40-ies, might still have close ties to the family members living in the country of origin, especially parents (or did the parent generation migrate along?). Stress about elderly parents living abroad, with limited possibility to go and visit can be a stressful event for migrants and can affect their family life during the time of the global pandemic. Does the data set contain any information that would allow disentangling of those mechanisms?

Response: Unfortunately, we did not collect the data that the reviewer suggested. For future studies, these migration-related variables should be examined in detail. We mentioned this possibility on page 24. 

“Seventh, stress about elderly parents living abroad, with limited possibilities for visitation, can serve as a stressful event for migrants and can affect their family life during global pandemics. In future studies, this possibility should be examined in detail.”

Comment 6: The authors use convenience sample – how representative is it for the whole migrant population? Is there a way to say anything about that?

Response: As the reviewer mentioned, we collected the data through convenience sampling strategies. We do not know how representative our sample may be vis-à-vis the migrant population in Israel as a whole. However, the sampling was conducted from all over Israel from different cities and settlements (we mentioned this on page 8). We briefly mentioned this possibility on page 24. 

“Eighth, although we sampled families throughout Israel, we collected the data through convenience sampling methods. We do not know how representative our sample may be vis- à-vis the population of migrant families in Israel.”

Comment 7: The conclusions about designing possible interventions to deal support migrant families in the time of the pandemic seem also very generic and applicable to general population. Again, what is so unique about this group, that does not apply to native population?

Response: Migrant families face increased challenges in adjusting to a new environment. These families are exposed to more stressors due to the changes in social and cultural norms. Therefore, they may require mental health services such as individual or family counseling, as well as help finding work following the pandemic.

---

## [Decision Letter · Decision Letter 1]

3 Nov 2022

Psychological distress, employment, and family functioning during the COVID-19 outbreak among recent immigrant families in Israel: Moderating roles of COVID-19 prevalence

PONE-D-22-04347R1

Dear Dr. Lee,

We’re pleased to inform you that your manuscript has been judged scientifically suitable for publication and will be formally accepted for publication once it meets all outstanding technical requirements.

Kind regards,

Ali B. Mahmoud, Ph.D.

Academic Editor

PLOS ONE

Additional Editor Comments (optional):

Reviewers' comments:

Reviewer's Responses to Questions

**Comments to the Author**

1. If the authors have adequately addressed your comments raised in a previous round of review and you feel that this manuscript is now acceptable for publication, you may indicate that here to bypass the “Comments to the Author” section, enter your conflict of interest statement in the “Confidential to Editor” section, and submit your "Accept" recommendation.

Reviewer #1: All comments have been addressed

2. Is the manuscript technically sound, and do the data support the conclusions?

Reviewer #1: Yes

3. Has the statistical analysis been performed appropriately and rigorously? 

Reviewer #1: Yes

4. Have the authors made all data underlying the findings in their manuscript fully available?

Reviewer #1: Yes

5. Is the manuscript presented in an intelligible fashion and written in standard English?

Reviewer #1: Yes

6. Review Comments to the Author

Reviewer #1: The authors have revised the manuscript according to the reviewers' instructions, the article is greatly improved, and I therefore consider it ready for publication

7. PLOS authors have the option to publish the peer review history of their article (what does this mean?). If published, this will include your full peer review and any attached files.

Reviewer #1: No

---

## [Editor Report · Acceptance letter]

7 Nov 2022

PONE-D-22-04347R1 

Psychological distress, employment, and family functioning during the COVID-19 outbreak among recent immigrant families in Israel: Moderating roles of COVID-19 prevalence 

Dear Dr. Lee:

I'm pleased to inform you that your manuscript has been deemed suitable for publication in PLOS ONE. Congratulations! Your manuscript is now with our production department. 

Kind regards, 

on behalf of

Dr. Ali B. Mahmoud 

Academic Editor

PLOS ONE